High expression of PIMREG predicts poor survival outcomes and is correlated with immune infiltrates in lung adenocarcinoma

Jiang Feng
Liang Min
Huang Xiaolu
Shi Wenjing
Wang Yumin wangyumin0577@wmu.edu.cn
Department of Laboratory Medicine, The First Affiliated Hospital of Wenzhou Medical University , Wenzhou, Zhejiang , China
Albertini Maria Cristina
Electronic publication date: 2021 Jul 6
Publication date: 2021
Volume: 9
Electronic Location ID: e11697
Received 2021 Jan 22; Accepted 2021 Jun 8
Copyright: ©2021 Jiang et al.
Copyright year: 2021
Copyright holder: Jiang et al.
License: This is an open access article distributed under the terms of the Creative Commons Attribution License, which permits unrestricted use, distribution, reproduction and adaptation in any medium and for any purpose provided that it is properly attributed. For attribution, the original author(s), title, publication source (PeerJ) and either DOI or URL of the article must be cited.
License URL: https://creativecommons.org/licenses/by/4.0/

Keywords: Prognosis, PIMREG, Immune infiltrate, Lung adenocarcinoma

Funding: National Natural Science Foundation of China 81672088 Zhejiang Provincial Natural Science Foundation LY19H200002 Wenzhou Municipal Science and Technology Bureau of China Y20170718 Zhejiang Provincial Research Center for Cancer Intelligent Diagnosis and Molecular Technology JBZX-202003 National Natural Science Foundation of China (81672088), Zhejiang Provincial Natural Science Foundation (LY19H200002), the Wenzhou Municipal Science and Technology Bureau of China (Y20170718), Zhejiang Provincial Research Center for Cancer Intelligent Diagnosis and Molecular Technology (JBZX-202003). The funders had no role in study design, data collection and analysis, decision to publish, or preparation of the manuscript.

==============================
Background

PIMREG is upregulated in multiple cancer types. However, the potential role of PIMREG in lung adenocarcinoma (LUAD) remains unclear. The present study aimed to explore its clinical significance in LUAD.

Methods

Using the Cancer Genome Atlas (TCGA) databases, we obtained 513 samples of LUAD and 59 normal samples from the Cancer Genome Atlas (TCGA) databases to analyze the relationship between PIMREG and LUAD. We used t and Chi-square tests to evaluate the level of expression of PIMREG and its clinical implication in LUAD. The prognostic value of PIMREG in LUAD was identified through the Kaplan–Meier method, Cox regression analysis, and nomogram. Gene set enrichment analysis (GSEA) and single-sample gene set enrichment analysis (ssGSEA) were performed to screen biological pathways and analyze the correlation of the immune infiltrating level with the expression of PIMREG in LUAD.

Results

PIMREG was highly expressed in patients with LUAD. Specifically, the level of PIMREG gradually increased from pathological stage I to IV. Further, we validated the higher expression of PIMREG expressed in LUAD cell lines. Moreover, PIMREG had a high diagnostic value, with an -AUC of 0.955. Kaplan–Meier survival and Cox regression analyses revealed that the high expression of PIMREG was independently associated with poor clinical outcomes. In our prognostic nomogram, the expression of PIMREG implied a significant prognostic value. Gene set enrichment analysis (GSEA) identified that the high expression PIMREG phenotype was involved in the mitotic cell cycle, mRNA splicing, DNA repair, Rho GTPase signaling, TP53 transcriptional regulation, and translation pathways. Next, we also explored the correlation of PIMREG and tumor-immune interactions and found a negative correlation between PIMREG and the immune infiltrating level of T cells, macrophages, B cells, dendritic cells (DCs) , and CD8+ T cells in LUAD.

Conclusions

High levels of PIMREG correlated with poor prognosis and immune infiltrates in LUAD.

Introduction

Lung cancer is the most commonly occurring cancer in men and the third most commonly occurring cancer in women, resulting in the highest lethality rates among all cancers worldwide (Bray et al., 2018; Siegel, Miller & Jemal, 2020). According to histopathological evaluations, 85% of lung cancer cases are non-small cell lung cancer (NSCLC) (Latimer, 2018). Of note, lung adenocarcinoma (LUAD) is the most common type of NSCLC (Chen et al., 2014; Meza et al., 2015). Despite significant breakthroughs in therapeutic interventions, such as surgery, chemotherapy, molecular targeted therapy, immunotherapy, and radiotherapy, the long-term outcomes of patients with LUAD at diagnosis remains unfavorable, and has been associated with a poor 5-y survival rate (AJW et al., 2019; Herbst, Morgensztern & Boshoff, 2018). To improve the diagnosis and prognosis of LUAD, the identification of biomarkers and novel immune-related therapeutic targets is urgently needed.

PIMREG, PICALM interacting mitotic regulator (FAM64A), was first described as a clathrin assembly lymphoid myeloid leukemia gene (CALM) affects the subcellular localization of the leukemogenic fusion protein CALM/AF10 (Archangelo et al., 2006). Furthermore, PIMREG has been reported to involve in the regulation of cell proliferation(Archangelo et al., 2008; Hashimoto et al., 2017; Hu et al., 2017; Zhang et al., 2014). In particular, PIMREG was shown to play a vital role in the enhancement of tumorigenesis and progression of neoplasms in multiple cancer types (Hu et al., 2017). Unfortunately, research on its function and the correlation of PIMREG with tumor immunity has not been reported in LUAD. Therefore, we first analyzed the transcription level and prognostic significance of PIMREG using data obtained from the Cancer Genome Atlas (TCGA) data. Moreover, we explored its biological mechanisms by GSEA analysis and further evaluated the association of PIMREG with the level of immune infiltration.

Materials and Methods

Patients and databases

The gene expression data of 513 LUAD tissues and 59 normal tissues were downloaded from the TCGA database. Then, we specifically sorted the data of 57 pairs of LUAD and adjacent normal lung tissues. and further screened the clinicopathological characteristics and prognostic data of the cases obtained from TCGA. Unified processing of RNAseq data in TPM format of TCGA and GTEx. The expression of PIMREG were analyzed via GTEx database, and TCGA database. the expression of PIMREG of datasets of Hou et al. (2010), Landi et al. (2008) and Selamat et al. (2012) were analyzed via the Oncomin database (https://www.oncomine.org). The HPA database was used to evaluate the protein expression of PIMREG in LUAD (Pontén et al., 2011).

Gene set enrichment analysis

We used GSEA to identify gene sets and pathways associated with PIMREG based on transcriptional sequences from obtained TCGA. In this study, gene expression data were divided into high- and low- expressing PIMREG groups. We performed GSEA to compare these two groups and identify potential functions via the Broad Institute Website using the R package cluster profiler (Subramanian et al., 2005; Yu et al., 2012).

Infiltration of immune cells

To assess the relative abundance of the tumor tissue-infiltrating immune cells, we performed ssGSEA (single-sample gene set enrichment analysis). We used the “GSVA” (R package) and immune data-sets, including 24 types immunocytes analyze the infiltration level of immune cells in LUAD expression profile data (Bindea et al., 2013). Spearman correlation and the Wilcoxon rank sum test were used to analyze the correlation between PIMREG and these different immune cells, as well as the association of immune cells with the high- and low-expressing PIMRE groups G.

Cell culture and quantitative real-time PCR

The A549 and H1299 lung adenocarcinoma cell lines were purchased from the Cell Bank of the Chinese Academy of Sciences. Cells were cultured in 1640 medium supplemented with 10% fetal bovine serum (GIBCO, USA) at 37 °C in a humidified CO2 incubator. The levels of expression levels of PIMREG were detected using an ABI 7500 Real-time PCR System. Relative expression levels were normalized to that of β-actin, which was used as an internal control. The primers used were as follows: PIMREG forward primer, 5′-CCTTAGTGGTGTCGGGGTCT-3′; reverse, 5′-GAGGTCCCCATGTTCTGCCA-3′. β-actin forward primer: 5′-CCTGGCACCCAGCACAAT-3′; reverse primer, 5′-GCTGATCCACATCTGCTGGAA-3′.

Statistical analysis

Statistical analysis of the expression of PIMREG in normal and LUAD groups was calculated using the Wilcoxon rank-sum test. We classified patients into two categories according to the “Median” expression of PIMREG. The clinicopathological features of PIMREG were analyzed using the Wilcoxon rank sum test or Kruskal-Wallis test and logistic regression. Prognostic analyses were performed using Kaplan–Meier survival analysis and Cox univariate and multivariate analyses. To evaluate the diagnostic significance of differentially expressed gene, a receiver operating characteristic (ROC) curve was generated using the “plotROC” package. The “rms” R package was adopted to plot the nomogram for the prognostic value among patients with LUAD. All statistical analyses were conducted in the R environment (v3.5.3) (http://www.r-project.org/). All figures were plotted using the R package ggplot2 (v3.1.0).

Results

Correlation analysis of expression of PIMREG in lung adenocarcinoma

Table 1 lists the expression of PIMREG and clinical data of 513 patients with LUAD downloaded from TCGA. We observed a significant relationship between the expression of PIMREG and clinicopathological parameters, such as T stage, N stage, pathologic stage, sex, smoking, tumor status, number of packs smoked per year, and age. In addition, we found that patients with high expression of PIMREG presented with elevated mutations in TP53 (p < 0.001) compared with those having a low expression of PIMREG (see Table 1).

Table 1 Correlation between PIMREG and clinicopathological characteristics.

Characters	Level	Low expression of PIMREG	High expression of PIMREG	p	
n		257	256		
T stage (%)	T1	102 (40.0%)	66 (25.9%)	0.003	
	T2	118 (46.3%)	158 (62.0%)		
	T3	25 (9.8%)	22 (8.6%)		
	T4	10 (3.9%)	9 (3.5%)		
N stage (%)	N0	180 (72.9%)	150 (59.1%)	0.005	
	N1	40 (16.2%)	55 (21.7%)		
	N2	26 (10.5%)	48 (18.9%)		
	N3	1 (0.4%)	1 (0.4%)		
M stage (%)	M0	172 (95.0%)	172 (91.5%)	0.252	
	M1	9 (5.0%)	16 (8.5%)		
Pathologic stage (%)	Stage I	156 (61.9%)	118 (46.6%)	0.006	
	Stage II	53 (21.0%)	68 (26.9%)		
	Stage III	33 (13.1%)	51 (20.2%)		
	Stage IV	10 (4.0%)	16 (6.3%)		
Primary therapy outcome (%)	CR	164 (74.9%)	151 (72.9%)	0.159	
	PD	28 (12.8%)	40 (19.3%)		
	PR	4 (1.8%)	2 (1.0%)		
	SD	23 (10.5%)	14 (6.8%)		
Gender (%)	Female	159 (61.9%)	117 (45.7%)	<0.001	
	Male	98 (38.1%)	139 (54.3%)		
Race (%)	Asian	5 (2.2%)	2 (0.9%)	0.657	
	Black or African American	26 (11.4%)	26 (11.9%)		
	White	197 (86.4%)	190 (87.2%)		
Residual tumor (%)	R0	162 (93.6%)	182 (96.8%)	0.063	
	R1	10 (5.8%)	3 (1.6%)		
	R2	1 (0.6%)	3 (1.6%)		
Anatomic neoplasm subdivision (%)	Left	97 (39.0%)	102 (41.0%)	0.714	
	Right	152 (61.0%)	147 (59.0%)		
Anatomic neoplasm subdivision2 (%)	Central Lung	29 (34.1%)	33 (31.7%)	0.848	
	Peripheral Lung	56 (65.9%)	71 (68.3%)		
Smoker (%)	No	50 (20.2%)	24 (9.6%)	0.001	
	Yes	198 (79.8%)	227 (90.4%)		
Tumor status (%)	Tumor free	158 (69.0%)	130 (56.8%)	0.009	
	With tumor	71 (31.0%)	99 (43.2%)		
TP53 status (%)	Mut	76 (29.8%)	165 (65.2%)	<0.001	
	WT	179 (70.2%)	88 (34.8%)		
KRAS status (%)	Mut	66 (25.9%)	73 (28.9%)	0.515	
	WT	189 (74.1%)	180 (71.1%)		
Age (mean (SD))	66.67 (10.01)	63.91 (9.88)	0.002	
number pack years smoked (mean (SD))	38.26 (27.02)	44.60 (27.03)	0.029	

Aberrant upregulation of PIMREG in lung adenocarcinoma

We used the TCGA database to reveal the mRNA expression pattern of PIMREG in LUAD tissues and normal tissues. We also assessed the expression levels of PIMREG in LUAD and normal tissues using Oncomine, GTEx, and HPA database. PIMREG expression was increased at the mRNA and protein levels (Fig. 1S). The average expression level of PIMREG mRNA was dramatically higher in LUAD tissues than in normal samples (p < 0.001 Fig. 1A). We then analyzed the expression of PIMREG in paired tumor and adjacent samples (Fig. 1B). We accordingly identified a significant difference, with tumor tissues exhibiting an increase in the expression of PIMREG. To evaluate the level of expression of PIMREG in various cancers, we performed a systematic analysis using data downloaded from TCGA databases. Our results showed that PIMREG was overexpressed in many tumor types, including bladder urothelial carcinoma, bone cancer, cholangiocarcinoma, and glioblastoma multiforme (Fig. 1C). Meanwhile, we observed that as the mRNA expression of PIMREG increased, the risk of pathological stage was also increased (Fig. 2A). In addition, we found that T stage (p < 0.001), N stage (p < 0.001), M stage (p = 0.065), TP53 status (p < 0.001), and smoking history (p < 0.001) were also significantly associated with the mRNA expression of PIMREG. To confirm the effect of the function of PIMREG in LUAD, we accessed the effect of the expression of PIMREG on the survival rate of patients with LUAD. Our results revealed that upregulation of PIMREG was linked to worse overall survival (OS), disease-specific survival (DSS), and progression-free interval (PFI) in patients with LUAD (Fig. 3). Moreover, we performed receiver operating characteristic (ROC) analysis of LUAD to reveal the diagnostic value of ABC, and found an area under the curve (AUC) value of 0.955.

Figure 1 The level of PIMREG expression in the mRNA based on TCGA database.

(A) Comparison of PIMREG expression level between LUAD tissues and normal tissue. (B) PIMREG expression level in 57 matched LUAD tissues and corresponding normal tissues. (C) Overview of PIMREG mRNA expression in different tumor tissues and adjacent normal tissues. BLCA, Bladder urothelial carcinoma; BRCA, Breast invasive carcinoma; CESC, Cervical squamous cell carcinoma and endocervical adenocarcinoma; CHOL, Cholangio carcinoma; COAD, Colon adenocarcinoma; ESCA, Esophageal carcinoma; GBM, Glioblastomamultiform; HNSC, Head and neck squamous cell carcinoma; KIRC, Kidney renal clear cell carcinoma; KIRP, Kidney renal papillary cell carcinoma; LIHC, Liver hepatocellular carcinoma; LUAD, lung adenocarcinoma ; LUSC, lung squamous cell carcinoma; PRAD, Prostate adenocarcinoma; READ, rectal adenocarcinoma; SARC, Sarcoma; STAD, Stomach adenocarcinoma; THCA, Thyroid carcinoma; UCEC, Uterine Corpus Endometrial Carcinoma.

Figure 2 Correlation between PIMREG expression and clinicopathological features.

(A) Clinical stage. (B) Tclassification. (C) N classification. (D) M classification. (E) TP53 status. (F) smoke.

Univariate and multivariate analyses of survival

To further explore the risk factors in patients with LUAD, we performed univariate and multivariate analyses. We noticed that the univariate analysis using the COX regression model showed that T stage, M stage, N stage, pathologic stage, primary therapy outcome, residual tumor and tumor status were associated with OS (p = 0.03, p < 0.001, p = 0.007, p < 0.001, p < 0.001, p < 0.001, p < 0.001, and p < 0.001, respectively) (Table 2). In addition, we detected that the expression of PIMREG (hazard ratio (HR) = 1.755, 95% confidence interval (CI) 1.304−2.361, Cox p < 0.001) was a significant predictor of OS. We then performed multivariate analysis, which revealed the independent risk factors. We found that the primary therapy outcome and tumor status were independent prognostic factors of OS inpatients with LUAD. However, we observed that the expression of PIMREG was still independently associated with OS (HR = 1.870, 95% CI [1.085–3.223], Cox p = 0.024). Furthermore, we validated the high expression of PIMREG in PFI and DSS (Tables S1, S2). Therefore, the high expression of PIMREG was demonstrated to be a good predictor of poor OS, DSS, and PFI in patients with LUAD.

Table 2 Univariate analysis and multivariate analysis of the correlation between clinicopathological characteristics and OS in LUAD.

Characteristics	Total (N)	HR (95% CI) Univariate analysis	P value Univariate analysis	HR (95% CI) Multivariate analysis	P value Multivariate analysis	
T stage (T2&T3&T4 vs. T1)	501	1.668 (1.184–2.349)	0.003	1.406 (0.737–2.685)	0.301	
N stage (N1&N2&N3 vs. N0)	492	2.606 (1.939–3.503)	<0.001	1.597 (0.691–3.693)	0.274	
M stage (M1 vs. M0)	360	2.111 (1.232–3.616)	0.007	1.023 (0.377–2.775)	0.964	
Pathologic stage (Stage II&Stage III&Stage IV vs. Stage I)	496	2.975 (2.188–4.045)	<0.001	0.748 (0.302–1.852)	0.53	
Primary therapy outcome (PD&SD&PR vs. CR)	419	2.818 (2.004–3.963)	<0.001	2.486 (1.442–4.286)	0.001	
Residual tumor (R1&R2 vs. R0)	352	3.973 (2.217–7.120)	<0.001	1.957 (0.723–5.294)	0.186	
Gender (Male vs. Female)	504	1.060 (0.792–1.418)	0.694			
Age (>65 vs. ≤65)	494	1.228 (0.915–1.649)	0.171			
Race (White vs. Asian&Black or African American)	446	1.422 (0.869–2.327)	0.162			
Anatomic neoplasm subdivision (Right vs. Left)	490	1.024(0.758–1.383)	0.878			
Anatomic neoplasm subdivision2 (Peripheral Lung vs. Central Lung)	182	0.913(0.570–1.463)	0.706			
number pack years smoked (≥40 vs. <40)	345	1.038(0.723–1.490)	0.84			
Smoker (Yes vs. No)	490	0.887(0.587–1.339)	0.568			
Tumor status (With tumor vs. Tumor free)	450	6.211(4.258–9.059)	<0.001	5.942(3.282–10.756)	<0.001	
TP53 status (Mut vs. WT)	499	1.254(0.936–1.680)	0.13			
KRAS status (Mut vs. WT)	499	1.087(0.779–1.517)	0.623			
PIMREG (High vs. Low)	504	1.755(1.304–2.361)	<0.001	1.870(1.085–3.223)	0.024	

Subgroup analysis

We further performed a subgroup analysis to assess the impact of the expression of PIMREG on OS according to age, sex, and anatomic neoplasm subdivision risk factors. We found that in each and every subgroup stratified by age, sex, and anatomic neoplasm subdivision, the high expression of PIMREG continued to lead to poor survival (Fig. 4).

Figure 3 The overall survival, progression-free survival and relapse-free survival rates in PIMREG high and low patient groups.

(A) Overall survival. (B) Disease-Specific Survival (DSS). (C) Progression-Free Interval (PFI). (D) ROC.

Figure 4 Subgroup analysis.

(A) The Kaplan–Meier curves for age ≤ 65 years subgroup. (B) The Kaplan–Meier curves for age ≥65 years subgroup. (C) The Kaplan–Meier curves for the male subgroup. (D) The Kaplan–Meier curves for the female subgroup. (E) The Kaplan–Meier curves for the left of anatomic neoplasm subdivision subgroup. (F) The Kaplan–Meier curves for the right of anatomic neoplasm subdivision subgroup.

Construction of nomogram

The above results suggested that the level of expression of PIMREG, primary therapy outcome and tumor status in patients with LUAD might be linked to prognosis. Therefore, we constructed a prognostic nomogram to predict individual survival probability through the levels of expression of PIMREG, primary therapy outcome and tumor status (Fig. 5A). The calibration curve of the prediction model showed that the established lines of 1-, 3- and 5-y survival matched the ideal line at a high degree (Fig. 5B). We observed that the C index of our prognostic nomogram reached 0.758 (0.734−0.782), indicating that the model had a reliable potential for the prediction of overall survival.

Figure 5 The role of PIMREG in predicting the prognosis of LUAD patients.

(A) Terminology diagram. (B) Calibration curve.

Gene set enrichment analysis identified identities an PIMREG-related signaling pathway

We next conducted GSEA to identify differences in the signaling pathways between low- and high- expression PIMREG data-set (Fig. 6). We respectively found that PIMREG was related to the mitotic cell cycle, mRNA splicing, DNA repair, Rho GTPase signaling, TP53 transcriptional regulation, translation pathways. Furthermore, GSEA revealed that the high level of PIMREG might be linked to cancer-promoting pathways.

Figure 6 GSEA analyses in LUAD patients with high expression of PIMREG compared with the ones with low expression.

NES, normalized enrichment score; ADJ, adjusted; FDR, false discovery rate.

Figure 7 The correlation between immune infiltrating level and PIMREG expression in LUAD.

Immune infiltration analysis

To understand the underlying cause of worse prognosis in patients with LUAD with immune infiltration, we carried out an enrichment analysis of the LUAD tumor microenvironment using ssGSEA. We found that the expression of PIMREG was negatively correlated with T cells, B cells, dendritic cells (DCs), macrophages, and CD8 + T cells (Fig. 7).

Correlation analysis of expression of PIMREG in lung adenocarcinoma

In order to evaluate the potential utility of PIMREG as a biomarker of LUAD, we further verified the expression of PIMREG in cell lines using qPCR (Fig. 8). We found that the mRNA level of PIMREG in the A549 and H1299 LUAD cell lines was increased 4.26- and 2.87-fold compared with that in 2B the normal lung epithelial cell line, which was consistent with our above-mentioned results.

Figure 8 The expression level of PIMREG in LUAD cell lines and normal lung epithelial cell line 2B.

Discussion

In this study, we found that the PIMREG gene was overexpressed in tumor compared with normal and adjacent normal tissues, and this overexpression was linked to worse OS, DSS and PFI. ROC analysis confirmed that PIMREG could be used as a biomarker for the prognosis of LUAD, while univariate and multivariate Cox analyses provided evidence that the mRNA expression of PIMREG might be an independent prognostic indicator of LUAD. Moreover, the level of expression of PIMREG was positivity correlated with the pathological stage. Patients with a high level of PIMREG were more likely to be presented with the disease in a late pathological stage, suggesting PIMREG as a tumor-related gene in LUAD. Furthermore, our prognostic nomogram exhibited satisfactory potential for clinical application.

Of note, the expression of PIMREG has been confirmed in several tumors (Archangelo et al., 2006; Barbutti et al., 2016). Using data obtained from TCGA databases, Jiao et al. showed that PIMREG was overexpressed in pancreatic cancer and related to poor outcomes; however no further experiments have been performed to verify this finding (Jiao et al., 2019). In addition, (Yamada et al. (2018) evaluated the prognostic value of the survival-associated gene PIMREG in breast cancer and renal cell carcinoma. Yao et al. also found that the expression of PIMREG was significantly associated with breast cancer, and patients with high expression of PIMREG were associated with poor prognosis. Further analysis validated this observation by demonstrating that PIMREG promoted the migration and proliferation of breast cancer cells (Yao et al., 2019). However, its level of expression and prognosis in LUAD have not been previously reported. Consistently, the current study demonstrated the high expression of PIMREG and its correlation with poor survival in patients with LUAD. In addition, we further found that the high expression of PIMREG continued to lead to poor survival even when patients where stratified according to age, sex, and anatomic neoplasm subdivision.

Prior studies have reported that higher expression of PIMREG promotes cell proliferation (Archangelo et al., 2008; Yao et al., 2019), whereas, silencing of PIMREG was reported to repress cell cycle-promoting genes in fetal cardiomyocytes (Hashimoto et al., 2017). Our results showed that the underlying molecular mechanisms through which ABC exerts its effect might be related to the cell cycle, mRNA splicing, DNA repair, Rho GTPase signaling, TP53 transcriptional regulation, and translation, as identified by GSEA.

In recent years, the role of immune cell infiltration in the development and progression of cancer has attracted increasing attention (Camidge, Doebele & Kerr, 2019; Carbone et al., 2015). Most studies have shown that infiltration of T and B cells in NSCLC predicts a favorable outcome (Edlund et al., 2019; Wang et al., 2019; Zhang, Endres & Kobold, 2019). High expression of CD8+ T cells has been demonstrated to predict increased survival in LUAD (Iglesia et al., 2016; Wang et al., 2020). In addition, macrophages have been reported to play an important role in regulating tumor innate and acquired immunity. For instance, the M1, CD204M2, and CD68 macrophages have been considered to confer protective immunity against several factors in the tumor microenvironment of NSCLC (Rakaee et al., 2019). However, macrophages are also known to promote tumor progression, Therefore, the role of the infiltration and activation of macrophages in cancer is not clear (Bercovici et al., 2019). Previous studies have also found that dendritic cells (DC) presented antigens to activate anti-tumor T cells (Wculek et al., 2020). Some studies demonstrated that the infiltration of DCs was related to protective immunity in LUAD(Wang, Huang & Li, 2019). In our study, we found that the expression of PIMREG was negatively correlated with T cells, macrophages, B cells, DCs and CD8+ T cells in LUAD, indicating that PIMREG might plays a significant role in regulating the tumor immune microenvironment. However, the mechanism by which PIMREG affects the tumor immune microenvironment and tumor progression in LUAD remains unclear. Further basic and clinical experiments are necessary to comprehensively elucidate the biological impact of PIMREG in lung cancer.

Conclusions

Overall, the high expression of PIMREG was correlated with prognostic implications. Moreover, PIMREG was a negatively correlated with T cells, B cells, macrophages, DCs and CD8+ T cells. Therefore, PIMREG, which might be associated with immune infiltration, might serve as a prognostic factor in patients with LUAD.

Supplemental Information

Supplemental Information 1 Univariate analysis and multivariate analysis of the correlation between clinicopathological characteristics and DSS in LUAD

Click here for additional data file.

Supplemental Information 2 Univariate analysis and multivariate analysis of the correlation between clinicopathological characteristics and PFI in LUAD

Click here for additional data file.

Supplemental Information 3 Raw data

Click here for additional data file.

Supplemental Information 4 Raw data of qPCR

Click here for additional data file.

Supplemental Information 5 MRNA and protein expression levels of PIMREG in LUAD

(A–C) PIMREG mRNA expression in normal lung tissues and LUAD tissues was detected using the Oncomine database. (D) PIMREG mRNA expression in normal lung tissues and paired LUAD tissues was analyzed based on TCGA and GTEx datasets. (E) HPA database were applied to analyze the characterization of PIMREG expression.

Click here for additional data file.

Additional Information and Declarations

Competing Interests

Author Contributions

Data Availability

The authors declare there are no competing interests.

Feng Jiang and Min Liang conceived and designed the experiments, analyzed the data, authored or reviewed drafts of the paper, and approved the final draft.

Xiaolu Huang and Wenjing Shi performed the experiments, prepared figures and/or tables, and approved the final draft.

Yumin Wang conceived and designed the experiments, authored or reviewed drafts of the paper, and approved the final draft.

The following information was supplied regarding data availability:

The raw measurements are available in the Supplemental Files.

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
