# Peer review of "High expression of PIMREG predicts poor survival outcomes and is correlated with immune infiltrates in lung adenocarcinoma"

_PeerJ, doi:10.7717/peerj.11697_

## Round 0.1 · original submission · Major Revisions

Please, address reviewer comments.

·

Basic reporting

The authors described PIMREG as a novel prognosis biomarker for LUAD patients.
The article is well written, the analysis seems to be well performed. There are some points to clarify concerning the text.

1. Please, re-written lines 32-33. It was not shown that PIMREG decreases immune cells, only a negative correlation was found.

2. In the Discussion section, line 167, it is written that "PIMREG could be used as a biomarker for the DIAGNOSIS of LUAD". Please re-write this sentence,
as data did not show PIMREG for diagnosis, but prognosis.

3. Please re-write the sentence in line 197. The word "perfect" seems not to be the better choice to describe a biomarker.

4. Please re-write the sentence in line 200. It is not clear macrophage role.

5. Discussion section lines 206-208. The negative correlation of immune cells and PIMREG expression. It is not clear why "PIMREG may enhance tumor activity of lung adenocarcinoma through these immune cells". Please clarify.

Experimental design

Major points

1. Only the TCGA-LUAD cohort was evaluated in the study. It is a good practice to confirm the biomarker validity in other cohorts, as many as possible.
For LUAD, there are other cohorts with RNA-Seq that can be found at the GDC data portal or GEO.

2. The cut-off methodology was not described in the Methods section (median, best cut-off?). Please describe the method in detail because it is of vital importance for results reproduction.

3. Although the authors found a negative correlation between PIMREG and immune cells, the coefficient rho was small
(never more than 0.5 ou -0.5). The correlations are not strong, even with a significant p-value which was influenced by the high number of patients in this study. These results should be taken more cautiously and another cohort should be
used for validation.

4. Concerning cell culture and quantitative RT-PCR, the authors used b2-actin as an internal control. Did they test another housekeeping gene?

5. Are there evidence for PIMREG at the protein level for LUAD patients? Could it be used as a biomarker? Is there any correlation between protein and mRNA level?

Validity of the findings

All data are available for reproduction. The conclusions are partially supported by the results because a validation cohort must be used.

·

Basic reporting

a. The language in the article is generally very clear and simple; a minor grammar check may be needed throughout the manuscript.
b. The article has good mentions of some previous work. However, the introduction needs to be enhanced significantly to include a more detailed review of the current understanding involving the role of PIMREG in cancers as well as background information on PIMREG itself.
c. Figure 4, Figure 6 font is too small. Overall resolution seems to be low, which makes it difficult to read figures effectively. Table formatting can be improved to enhance readability. Figure 1C acronyms need to be displayed and spelled out at least in SI.
d. Results relevant to the presented hypothesis are appropriately included.

Experimental design

a. The article is well within the scope of the journal. The article clearly defines the overall research goal. It would be helpful to more clearly define the specific research questions earlier on in the manuscript. The experiments are proposed in attempt to achieve the general goal of the research.
b. In general, the investigations are carried out rigorously. A brief mention of the ethical guidelines from human patient database use would be helpful to include in the methods.
c. The materials and methods section provides a good overview of the analyses performed including the various statistical tests.

Validity of the findings

a. In general, the data presented in the manuscript represents a solid base for the conclusions. Statistical analyses are robust and used appropriately.
b. The conclusions from the studies are concisely stated. Expanding conclusions to include discussions of study limitations would be helpful.

Additional comments

1. Line 101 “There is a significant relationship between the PIMREG expression with the clinicopathological parameters such as T stage, N stage, Pathologic stage, gender, smoke, tumor status, number pack years smoked and age.” This statement is too general and does not seem to be backed by or referenced to the data. Such unreferenced claims need to be checked throughout the introduction and discussion.
2. Fig 8, more details on the expression measurement should be provided (e.g. fold change, relative levels, etc.)
3. More details on the choice of ADH1C primer in mRNA expression experiments are needed. Definitions needed for that gene and why it is used to detect PIMREG

---

## Round 0.2 · Minor Revisions

A few more adjustments are needed as suggested by the Reviewer.

·

Basic reporting

The authors made changes and improve manuscript quality. PIMREG increased expression on other cohorts corroborates with the manuscript data.
Also, protein expression seems to agree with mRNA data, although only a representative picture was depicted.
There are a few misspelling sentences, mainly in the Methods section to be adjusted.
In the Discussion section, the authors claimed "Our study also demonstrated that PIMREG was correlated with immune infiltration
in LUAD, which might be the mechanism by which PIMREG affects the prognosis of LUAD."
Then, the authors rewrote the discussion in the last paragraph contradicting this sentence. Please, adjust the text.

Experimental design

The experimental design is suitable for the manuscript.

Validity of the findings

The manuscript is clear and the results sustain the conclusions.

---

## Round 0.3 · accepted · Accept

As suggested by Reviewers, the article is adequate and no further revision is needed.

·

Basic reporting

The article was revised adequately.

Experimental design

The experimental design was evaluated in the previous review steps.

Validity of the findings

The results are clear.

Additional comments

The article is adequate and no further revision is needed.